# Lipid Nanoparticles as Promising Carriers for mRNA Vaccines for Viral Lung Infections

**DOI:** 10.3390/pharmaceutics15041127

**Published:** 2023-04-03

**Authors:** Mena Hajiaghapour Asr, Fatemeh Dayani, Fatemeh Saedi Segherloo, Ali Kamedi, Andrew O’ Neill, Ronan MacLoughlin, Mohammad Doroudian

**Affiliations:** 1Department of Cell and Molecular Sciences, Faculty of Biological Sciences, Kharazmi University, Tehran 1571914911, Iran; 2Department of Clinical Medicine, Tallaght University Hospital, Trinity College Dublin, D02 PN40 Dublin, Ireland; 3School of Pharmacy and Pharmaceutical Sciences, Trinity College Dublin, D02 PN40 Dublin, Ireland; 4Research and Development, Science and Emerging Technologies, Aerogen Limited, Galway Business Park, H91 HE94 Galway, Ireland; 5School of Pharmacy & Biomolecular Sciences, Royal College of Surgeons in Ireland, D02 YN77 Dublin, Ireland

**Keywords:** nanoparticles, LNP, mRNA vaccines, viral infections, delivery systems, respiratory pathogens

## Abstract

In recent years, there has been an increase in deaths due to infectious diseases, most notably in the context of viral respiratory pathogens. Consequently, the focus has shifted in the search for new therapies, with attention being drawn to the use of nanoparticles in mRNA vaccines for targeted delivery to improve the efficacy of these vaccines. Notably, mRNA vaccine technologies denote as a new era in vaccination due to their rapid, potentially inexpensive, and scalable development. Although they do not pose a risk of integration into the genome and are not produced from infectious elements, they do pose challenges, including exposing naked mRNAs to extracellular endonucleases. Therefore, with the development of nanotechnology, we can further improve their efficacy. Nanoparticles, with their nanometer dimensions, move more freely in the body and, due to their small size, have unique physical and chemical properties. The best candidates for vaccine mRNA transfer are lipid nanoparticles (LNPs), which are stable and biocompatible and contain four components: cationic lipids, ionizable lipids, polyethylene glycols (PEGs), and cholesterol, which are used to facilitate cytoplasmic mRNA delivery. In this article, the components and delivery system of mRNA-LNP vaccines against viral lung infections such as influenza, coronavirus, and respiratory syncytial virus are reviewed. Moreover, we provide a succinct overview of current challenges and potential future directions in the field.

## 1. Introduction

Due to the inherent characteristics of viral disease, which include complexity in the life cycle, different stages of proliferation in different chambers or subcellular organs, differences in replication dynamics, the possibility of latent infection, and drug resistance, it has become necessary to develop new treatment modalities in the context of infection [1,2]. It is important to understand the limitations, such as the stability of therapeutic agents within the cellular environment. A reduction in drug efficacy may occur, for example, due to the instability of the compound within the cell, due to changes in the genetic composition of the receptors at the cellular level or due to changes in intracellular signaling pathways due to disease progression [3,4]. Recently, there has been a great deal of interest in RNA-based technologies for the development of vaccines for the prevention and treatment of disease [5]. As such, mRNA vaccines have made great strides in recent decades and have become an invaluable solution for infection prevention and treatment, most notably in the context of SARS-CoV-2 infection [6,7,8]. mRNA vaccines can be manufactured at scale, faster than other common vaccines, and can be developed quickly using targets from the viral genome [9]. The delivery of mRNA vaccines is possible in different ways, such as encapsulation by delivery carriers, peptides, free mRNA in solution, lipid nanoparticles, and polymers, and ex vivo through dendritic cells [8]. mRNA technology has demonstrated significant potential in clinical applications by preventing infectious disease and cancer [10]. Due to limited licenses and access to adjuvants for human antiviral vaccines, nanoparticles have been identified as promising adjuvant candidates for use in vaccines against various viruses [11]. In the emerging field of nanomedicine, materials are used that have nanoscale dimensions to diagnose or treat several diseases. Additionally, due to their unique physical properties, these materials are involved in the targeted delivery of drugs to specific locations around the body [12,13,14]. One of the most notable benefits of nanoparticles in the context of the improved targeted delivery of therapeutics is that they aid in the evasion of numerous deleterious factors, for example, endosomal acidification, which decreases drug concentration and payload at target sites. Through the evasion of these factors, nanoparticles act to increase drug efficacy [15,16]. Furthermore, it is also important to note that nanoparticles and nanotechnology can provide new solutions to some problems in traditional medicines and vaccines, such as insolubility in water, sensitivity, and absorption [17,18,19,20]. Nanoparticles have already been shown to protect encapsulated mRNA and help increase its stability. As such, the presence of these suitable carriers can prevent degradation and enhance immunity [21,22]. In order to better understand the effectiveness of this treatment platform, below, we explore a variety of delivery methods and systems.

## 2. mRNA Vaccination and Mechanism of Action of mRNA

Vaccination is the most successful way to prevent and control disease, and it has saved countless lives in both infectious disease and cancer treatment [5,22,23]. At present, nucleic acid vaccines are divided into two categories: plasmid DNA vaccines and mRNA vaccines, both of which have demonstrated potential [22]. As the mRNA vaccines developed from the first generation of vaccines are attenuated vaccines, they are called third-generation nucleic acid vaccines, which do not contain live virus and do not pose an infection risk to vaccinated persons [22]. Instead, these vaccines contain a strand of genetic material called mRNA inside a special coating [24]. mRNA vaccines provide a strong immune response involving cytotoxic antibodies and T cells, which results in an effective, safe, and long-lasting response [5]. The purpose of these vaccines is to transfer RNA to produce protein antigens to induce the immune response and expression in the target cell [22,24,25].

In this way, parts of the genome are copied and mRNA is transferred to the cytoplasm. Once in the cytoplasm, the mRNA is read and instructs cells on what protein to make and how, resulting in the triggering of an immune response in the body against this antigen. The immune system is activated and stores these antigens and fights them if proteins similar to the foreign substance appear [25]. mRNA is transiently active and degrades through metabolic pathways, so it does not act on the host for homeostasis and can meet all the genetic information needs to encode and express a variety of proteins. Through modifying the mRNA sequence, vaccine production can be optimized. Although encoded antigens are different, most mRNA vaccine production and purification processes are quite similar, so it can be maintained or standardized to produce other similar mRNA vaccines [26]. mRNA vaccines must also activate innate immunity. By stimulating inflammatory stimuli, they initiate the lymphatic migration of innate immune cells and activate antigen-presenting cells, which in turn stimulate the production of cytokine signals and responses [27]. mRNA technologies have the potential to revolutionize the medical field, including the prevention of infectious diseases; this is a research area that is currently under intense development [10,28].

### 2.1. Background of mRNA Vaccines

mRNA was first discovered in 1961 by Brenner et al., but it was not until 1989 when the concept of mRNA-based drugs was imagined, when Malone et al. demonstrated that mRNA can be transferred [26]. Following the development of an extensive transmission method in vitro, the use of mRNA as a treatment of modality was supported for the first time [29]. One of the most promising alternatives to conventional vaccine methods was nucleic acid therapy. The first report of the use of in vitro-transcribed (IVT) mRNA in animals was published in 1990 (Figure 1). The injection of IVT mRNA in mice resulted in the local production of the encoded reporter protein and the induction of an immune response against the selected antigen [26,28,30]. The discovery, development, and improvement in new vaccines has been made possible by the use of new technologies, increasing the strength of the immune system [30].

### 2.2. Therapeutic Considerations of mRNA Vaccines

In recent years, there has been a great deal of focus on the use of mRNA, with advances leading to the development of vaccines, cancer immunotherapies, protein therapy alternatives, and genetic therapies [22,31,32,33]. Some of the advantages and disadvantages of mRNA vaccines are mentioned in Figure 2. These vaccines have many advantages over plasmid DNA (pDNA) and vaccines based on viral vectors. Notably, these vaccines do not integrate into the genome of host cells and do not alter the genome, and since recombination between single-stranded RNA is rare, cytosolic mRNA has no interaction with the genome. It can retain the properties of DNA vaccines and is very effective and clinically safe. They are delivered for on-site antigen expression without the need to cross the nuclear membrane barrier for protein expression, and can express complex antigens without packaging restrictions, so they are readily available [6,10,22,27,34,35,36,37]. As discussed, one of the main advantages of mRNA vaccines is the speed at which they can be produced. The synthesis and purification of mRNA is faster, easier, and less expensive than other vaccines [9]. All these factors combine to give mRNA vaccines a significant advantage in the rapid response to new epidemics and pandemics.

mRNA vaccines save both time and costs, and are produced quickly after the immunogen-encoding sequence is available, a process that is cell-free and scalable. In vitro mRNA vaccines are easy to produce, and in vivo expression prevents protein and virus contamination [6,26,38,39]. mRNA production avoids common risks associated with other vaccine platforms, such as live viruses, viral vectors, inactivated viruses, and subunit protein vaccines. In fact, the process of making mRNA does not require toxic chemicals or cell cultures that can be infected with adventitious viruses [28]. The risks of genome integration, long-term expression, and induction of autoantibodies to anti-DNA pathogens have prevented the Food and Drug Administration (FDA) from approving DNA-based vaccines for human use. The human use of viral vectors is a threat because of their potential return to pathogenicity and the presence of strong carrier-specific humoral immune responses, especially on amplification. As these concerns do not apply to mRNA vaccines, they are therefore not classified as “gene therapy” by the FDA [40]. Because vaccines are administered to healthy people, the safety requirements for modern preventive vaccines are very strict. As a result, people at risk for increased autoimmune reactions need to take reasonable precautions before mRNA vaccination [28].

Another potential safety issue is the presence of extracellular RNA during mRNA vaccination, which has been shown to increase the naked cell extracellular RNA permeability of endothelial cells and may, therefore, cause oedema [28]. Other studies have shown that extracellular RNA causes blood coagulation, resulting in pathological thrombosis [28]. As different mRNA methods and delivery systems are being used for the first time in humans and are being tested in larger patient populations, safety requirements are being further assessed on an ongoing basis [28].

mRNA-based therapies have numerous delivery requirements to ensure mRNA stabilization under physiological conditions [32,39]. The main challenge for the development of mRNA in vaccines lies in the optimization of stability and delivery systems due to the instability and easy degradation of mRNA molecules [22,41]. Although some of these challenges have been partially addressed through mRNA modification, intracellular mRNA delivery is still a major hurdle [32]. However, there is a wide field for the further development and improvement in mRNA-based vaccines [29].

## 3. Combining Nanoparticles with mRNA Vaccines

To optimize the delivery systems of standardized mRNA vaccines, a number of key factors should be considered. The mRNA encoding the antigen must be well packaged and protected; this increases the thermal stability of the vaccine to allow its storage and transport at room temperature. Furthermore, this has adjuvant properties to effectively induce cellular and humoral immune responses to a wide range of pathogens. The effective protection of the mRNA ultimately functions to boost efficacy and enable the practical production and distribution of these vaccines in a cost-effective manner [42].

The development of nanoparticles in recent decades has led to their use to overcome biological barriers from systemic to cellular levels and to reduce the limitations of free drug molecules. Nanoparticles improve the solubility of drugs, aid in treatment escape from the immune system, and increase the half-life of drugs in the circulatory system. In addition, nanoparticles can deliver drug compounds to work together in the target cell and program to accurately absorb specific drugs, as well as releasing them into the target cell [43]. mRNA has a general negative charge, its molecular size is large, and it is hydrophilic. On the other hand, the cell membrane is lipid-friendly and negatively charged, so mRNA tries to pass through it through diffusion. As this mRNA is foreign, it is recognized by the host as foreign matter, so ribonucleases (RNases) cleave it into small nucleotide fragments. This cleavage causes only a small amount of mRNA to pass through the membrane subdomain. It is rich in lipid rafts that attach to the inner cell membrane, resulting in the accumulation of lysosomes, where most mRNA is decomposed. Nanotechnology and the use of nanoparticles have solved some problems of intracellular delivery of mRNA and offer protection from these processes [44,45].

Nanoparticle-based vaccines have the ability to interact with antigen-presenting cells (APCs). APCs are specific cells that, in some cases, can be normal cells and can produce antigens that increase the immunity of T cells. These vaccines increase the level of antigen produced by APCs. Furthermore, they can also mimic pathogens and increase the production of multivalent antigens by APCs to improve and increase immunity. This process and these features have led scientists to develop nanoparticles for use in the treatment of and vaccination against several diseases [43]. In nanoscience, there are technologies for controlling and manipulating nanosized objects, e.g., “nanoparticles”, which have special functions and have significant differences in nanometer dimensions with other identical materials [46]. Nanoparticles have created new solutions in medical applications due to their interaction with complex organisms and cellular mechanisms [14,46]. Nanoparticles are used to load antigens and prevent degradation and prolonged exposure to antigens. Due to their characteristic physical and chemical properties, they deliver antigens to specific locations and induce a variety of immune responses [11].

## 4. Nanoparticles as Carriers

To introduce nanoparticles that can be used as carriers, we first must consider size and appearance. Typically, they have a wide range of sizes between 100 and 500 nm, which are converted into smart systems through manipulation of the size and properties of the surface and material used [47,48,49]. The use of nanoparticles for therapeutic functions has advantages. The size of nanocarriers affects the circulation time due to their small size. These particles have a large surface-to-volume ratio, which makes them more soluble. Additionally, their surface charge can be adjusted for easier cell entry into the negatively charged cell membrane [14,50]. Due to the importance of biological structure and organisms, safe and compatible nanocarriers should be biocompatible—that is to say, having the ability to integrate with a biological system without causing an immune response or any adverse effect [51]. Furthermore, they should be nontoxic [52]. Coupled with hydrodynamic size, shape, amount, surface chemistry, route of administration, and immune system response, these are the key considerations in the context of a successful nanodrug delivery system [52,53]. Nanoparticles, due to their size at the atomic or molecular level, move more freely in the human body than larger particles [54,55]. Studies have shown that nanoparticles with a hydromechanical diameter of 10 to 100 nm have better pharmacokinetic properties in vivo, and the smaller the nanoparticle, the better the tissue expansion and renal clearance, and the larger the nanoparticle, the faster it is opsonized. It is released from the bloodstream with the help of macrophages [53]. The exploitation of these characteristics allows nanoparticles to be used to deliver genes, vaccines, hydrophobic drugs, proteins, and other large biomolecules. [56,57].

### Nanoparticle Delivery Systems

Nanoparticle drug delivery systems must have certain characteristics and principles in order to achieve targeted delivery, so they must first be able to reach specific target tissues, be able to identify them, and deliver the drug, all the while preventing or reducing drug-induced damage to healthy tissue [49]. The controlled release of drugs from nanocarriers is possible in several ways: changes in physiological surroundings inclusive of enzymatic activity, pH, temperature, or osmolality [53]. Generally, cell-specific targeting with nanocarriers, i.e., the penetration of particles into inflamed tissue by larger epithelial junctions, is divided into active and inactive categories. In active nanocarriers, the drug delivery system conjugates to a selected cell ligand. In passive systems, the nanoparticle reaches the target organ due to leaky connections [13,49,53]. The dependence of each drug delivery system on different drug polarities depends on the chemical and physical properties and morphology of that system and can be changed through chemical reactions such as hydrogen and covalent bonds or through physical reactions such as Van der Waals reactions or electrostatic interactions [54]. For targeted treatment, it is necessary to pay attention to how the drug binds to the nanocarrier because there are different methods for different drugs—these include adsorption, covalent bonding, and encapsulation. The advantage of covalent bonding is the ability to control the number of molecules, whereby we can control the amount of therapeutic compound delivered [53]. In general, nanoparticle delivery systems include both nanocapsules and nanospheres. To define these two terms, it should be noted that nanocapsules are vesicular systems in which the drug is enclosed in a cavity surrounded by a polymer membrane, while nanospheres are matrix systems in which the drug is evenly distributed. Their generation depends on the methodology used during the preparation of the products, with both having different respective release properties of their contents [46]. The mechanisms used to release drugs from nanocarriers include solvent diffusion and chemical reactions, which release contents in a controlled manner. Developments in the field have led to the generation of strategies to improve the status of nanocarriers, which in some cases do not show hybrid affinity with modified ligands [54]. The process of drug encapsulation begins with drug dissolution, followed by entrapment, absorption, and finally binding or enclosure within or on the nanomatrix [46].

## 5. Lipid Nanoparticles Used in mRNA Vaccines and Their Delivery Systems

Nanoplatforms that can be used to deliver mRNA in the body include polymer-based nanoparticles, polymer–lipid hybrid nanoparticles, polypeptides, protein derivatives, complexes of protein mRNA derivatives, and gold nanoparticles. All of these have the relative ease of large-scale production, scaling their manufacturing ability, to provide the ability to deliver enclosed mRNA delivery to ribosome mRNA (via endosome escape), and flexibility is used to modify the surface with a ligand to further target specific cells [44,58]. Lipids are the best choice for mRNA transfer due to their fusion compatibility with lipid cell membranes [59]. Some advancements in lipid nanoparticles are listed in Figure 3. First, to identify lipids, it must be noted that these are amphiphilic molecules that consist of a hydrophobic tail region, a polar head group, and a linker between the two groups [60]. LNPs are stable, biocompatible particles with a bilayer lipid shell consisting of four components: cationic/ionizable lipids, PEG, auxiliary lipids, and cholesterol, that encompass an aqueous core in which they can carry mRNA to protect it from destruction (Figure 4) [22,59,61]. The ratio of these components varies based on the target tissue, and are differentiated from liposomes due to the lack of a hollow nucleus [59]. It can be noted that the function of liposomes in drug delivery is absorption. Liposomes are also spherical lipid vesicles but have two concentric lipid layers and reach one or more layers that are confined to a water space (Figure 4). Liposomes have the ability to carry hydrophobic and hydrophilic molecules. Cationic liposomes, which are mainly composed of cationic lipids, can effectively concentrate nucleic acids due to positive charges [56,62]. The comparison between liposomes and LNPs is shown in Table 1. The use of liposomes in mRNA delivery can have the advantage that they can protect mRNA against nucleases, they have high transfer efficiencies, and because of their resemblance to cell membranes, they can easily combine with receptor cells. As they are not host constrained, they can simulate cell membranes to achieve long-term storage stability [61]. However, the downside of liposomes is that, because of their low encapsulation efficiency of oligonucleotides, they cannot be used for the in vivo development and delivery of nucleic acid vaccines [59]. The selection of LNPs for mRNA vaccine delivery has changed from the past, from lipoplexes, cationic nanoemulsions, and traditional liposomes to more advanced solid nanoparticles and nanostructured lipid carriers [44,59].

### 5.1. Ionizable Lipids

Ionizable lipids are considered the most important component of LNPs. These lipids normally protonate in the endosome at pH 5 to have a positive charge. This positive charge can form ion pairs in the endosome membrane with the help of negatively charged phospholipids, disrupting the bilayer structure. It destabilizes membranes and facilitates the endosomal escape of nanoparticles by helping them fuse with endosome membranes and releasing them into the cytoplasm. However, ionizable lipids that can be used in vaccines to deliver mRNA in vivo remain neutral at physiological pH to ultimately reduce positive charge-related toxicity in terms of safety and tolerability characteristics after vaccination. As neutral lipids have fewer interactions with the anionic membrane of blood cells, they improve the biocompatibility of lipid nanoparticles. mRNA ionizable lipids are formulated into nanoparticles in an acid buffer so that the lipids have a positive charge and absorb the RNA charge [6,36,44,59,60,63,64,65]. Notably, ionizable lipids have been shown to be the main stimulants of immunogenicity, pharmacokinetics of LNPs, and tissue tolerance, because the more biodegradable the ionizable lipids, the faster the association between protein expression and immunogenicity. Muscles, the spleen, and the liver are targeted without necessarily compromising immune responses. In addition, biodegradable ionizable fats are removed more quickly from the injection site, so they have less tissue stimulation. Note, however, that depending on the lipid structure, some positively charged lipids can act as vaccine auxiliaries and stimulate the innate immune response [44].

### 5.2. Cationic Lipids

Cationic lipids are amphiphilic molecules that are made up of three parts and have a group of amines that produce a pure and permanent positive charge. In addition, they have a hydrophobic chain and a bonding group that allows the hydrophilic part to be attached to the hydrophobic chain. Their positive charge makes them easy to combine with negatively charged nucleic acids, which helps improve delivery and drug delivery efficiency. Additionally, their positive charge increases cellular uptake and endosomal escape of the LNP system by disrupting the cell/endosomal membrane. In other words, they form a nucleus during this event that, when it reaches solubility, other lipids attach to, which increases the release of endosome mRNA into the cytoplasm [59].

The highest proportions included in the structure of LNPs are cationic lipids/ionizers, which are included in the highest ratio. Although these lipids are effective in mRNA delivery, they also stimulate pro-apoptotic and pro-inflammatory toxic responses. The determinant of mRNA transfer efficiency is the ionizable cationic phospholipid, which is critical and must be ionized under physiological and acidic conditions to improve transmission efficiency [66]. For this to happen, the tertiary amine must be protonated, and phospholipids form a hydrophobic chain tail and a smaller geothermal ion head to form a conical structure and turn the membrane into a hexagonal crystalline phase. The accumulation of phospholipids in the body may have side effects and toxicity; to counteract this, carboxylic ester is present to ensure the degradability of phospholipids in the body [61]. For example, 1,2-di-O-octadecenyl-3-trimethylammonium-propane (DOTMA), a quaternary ammonium lipid, is marketed as Lipofectin in combination with 1,2-dioleoyl. sn-glycero-3-phosphoethanolamine (DOPE)—dioleol-3-trimethyl ammonium-propane (DOTAP) is a biodegradable analogue of DOTMA. All of these have been used to deliver mRNA to cell types [6,59,60,61,63]. Given the introduction of both classes of lipids, for comparison, we can say that in some cases, ionizable lipids used as components of lipid mRNA nanoparticles may be significantly more expensive than the cationic lipids used in the use of new ionizable lipids, though less clinical data are available [59,60].

### 5.3. Other Types of Lipids That Can Be Used in mRNA Delivery Systems Based on Nanoparticles

Other lipid–mRNA nanoparticle formulations typically contain components such as phospholipids, for example, phosphatidylcholine and phosphatidylethanolamine, PEG- functionalized lipids (PEG-lipids), or cholesterol. In general, these lipids can improve the properties of nanoparticles such as transfer efficiency, particle stability, tolerability, and biodistribution, as well as increase cellular uptake and lipid bilayer instability, thereby improving nucleic acid transport efficiency [44,60,67]. Choosing an optimal auxiliary lipid depends on the ionizable lipid material and the RNA load. Auxiliary lipids increase efficiency by promoting lipid phase transfer, which promotes membrane fusion with the endosome and modulates the fluidity of nanoparticles. Auxiliary phospholipids are neutral and are usually saturated phospholipids that improve overall stability and phase transition temperature. The combination of phospholipids is important for its ability to support the lipid bilayer structure. To refer to examples of this type of lipid, the following can be mentioned: the distearoyl-snglycero-3-phosphocholine (DSPC) structure used in mRNA-1273 and BNT162b2 COVID-19 vaccines is a saturated tail phosphatidylcholine that allows DSPC molecules to form a layered phase that causes phase stabilization. DOPE is also a phosphoethanolamine with two unsaturated tails that destabilizes endosomal membranes and facilitates the endosomal escape of lipid nanoparticles [6,36,60,61,63].

### 5.4. Lipid-Bound Polyethylene Glycol (PEG)

PEG lipid consists of a hydrophilic PEG polymer conjugated to a lipid. Interestingly, PEGylated lipids are usually included in mRNA-LNPs at lower molar ratios than other components of the formulation. PEG lipids are key factors in controlling particle size so that, by increasing the ratio of a PEG lipid in the LNP formulation, they increase the compression in the lipid bilayer and the repulsive forces between the particles that can produce smaller particles. PEGylated phospholipids are present on the surface, which increases the hydrophilicity of the compound. This component prevents rapid clearance by the immune system and increases stability by preventing the accumulation of particles [22,61,68]. They also prevent nonspecific binding to proteins (opsonins) and are eliminated by the reticuloendothelial system in the body. PEG lipids and some PEG modifications extend the circulation time of nanoparticles by reducing the clearance by the kidneys and the mononuclear phagocytic system (MPS). PEG lipids are used to conjugate specific ligands to the particle for targeted delivery [36,44,60].

It should be noted that the optimization of the PEG lipid component density and chain length is very important for the effective development of mRNA-LNP systems. By adjusting the molecular weight of PEG and the length of the lipid anchor, the efficiency, circulation time, and absorption of immune cells can be changed, depending on the application. This is because the presence of a hydrophilic layer around LNPs can interfere with cell uptake, limiting endosome instability and thus reducing transfection power. There is a PEG dilemma in that, although faster separation of PEG lipid from the LNP surface reduces particle circulation time, it can increase its synthesis and allow the LNP to deliver its charge more efficiently to target cells, which can adversely affect transmission efficiency [59,63,69].

The amount of coating on lipid nanoparticles should be appropriate because if this amount is more than necessary, it reduces the rate of cell uptake and interaction with the endosomal membrane, and more PEG lipids increase the circulation time of lipid nanoparticles. Research and reports have shown that PEG-lipid coatings significantly increase lymphatic drainage. However, we know that this response does not necessarily produce a stronger immune response and may increase the protection of cationic charges of lipid nanoparticles against interactions with nonspecific proteins. Additionally, in cases of repeated intravenous administration of these nanoparticles, the response of anti-PEG antibodies causes severe sensitization by accelerating the blood passage of lipid nanoparticles. These reported results are important for a variety of immunotherapy applications [36,44,59,60,61,63,65].

### 5.5. Cholesterol

Cholesterol is a natural component of cell membranes and plays an important role in reducing the transfer temperature of LNPs, facilitating the transfer from the layered phase to the hexagonal phase, helping integrate these lipid nanoparticles with the endosome, and finally releasing mRNA from the LNPs internalized into the cytosol [61]. Cell cholesterol shows little solubility in the nanoparticle core, so it may accumulate crystalline on the surface of LNPs, causing lipid bilayer instability and increasing endosomal escape. In addition, it has a strong membrane fusion ability that is responsible for the intracellular uptake of mRNA and cytoplasmic entry [61]. Cholesterol is also a particle stabilizer in LNP formulations that limits nonspecific LNP protein interactions. It improves fat packaging and modulates the membrane fluidity and permeability of the bilayer membrane system through a condensing effect. Cholesterol also increases the ability of the phospholipid membrane to form a bicontinuous cubic phase at physiological temperatures by mixing with unsaturated phospholipids in a fat transfer system, thereby further increasing melting [36,44,59,61,64,67]. The purpose of lipid nanoparticle–mRNA systems is determined by reaching the target tissues because the target cells must then be internalized, and finally, the mRNA molecules must produce the necessary proteins which must escape the endosomes to reach the cytoplasm somewhere where translation happens [60]. The details of lipid nanoparticle delivery systems for mRNA-based vaccines are noted in Table 2.

## 6. Nanodelivery Systems Based on Lipid Nanoparticles

It is preferable to deliver mRNA based on lipid nanoparticles through endocytosis and then by electrostatic binding, as well as through fusion with the cell membrane through reverse non-bilayer lipid phases. This is because they play a role in protecting the mRNA of lipid nanoparticles, which increases cellular uptake and thus increases endosomal escape to facilitate cytoplasmic delivery (Figure 5) [63]. So far, we have described the many benefits of using lipid nanoparticles in the context of effective vaccination due to their ability to train the immune system [63]. 

In delivering mRNA through nanoparticles, LNPs have two important roles:(1)Promotion of the release of mRNA from the endosome to the cytoplasm;(2)Control of the uptake of mRNA into the target host cell.

As a result, they can provide mRNA bioavailability to ribosomes, in which it produces antigen mRNA, and as mentioned, LNPs can protect mRNA from being destroyed by extracellular RNases by encapsulating them within the nucleus. Another responsibility of LNPs is to improve cellular uptake through various endocytosis pathways, so the post-endocytic stage is when LNPs are trapped within the endosome. Despite all of the above, unknown mechanisms still exist and further research is required [44]. The presence of LNPs is useful in delivering mRNA to intracellular ribosomes. After the intramuscular injection of LNP vaccines, extensive local tissue exposure to the injection site is provided by LNPs. Of course, the pharmacokinetics of these vaccines depend on several factors, including surface charge, particle size, the colloidal stability of LNPs, and the biodegradability of cationic or ionizable lipids. LNPs less than 150 nm in size have been shown to be depleted through afferent lymph vessels. Larger LNPs are readily delivered by phagocytic immune cells and then to the lymph nodes [36,44].

## 7. Treatment and Prevention of Viral Infectious Diseases of the Lungs

To date, significant effort has been focused on the development of vaccines against viral infections such as influenza, SARS-CoV-2, MERS-CoV, and RSV, which are associated with a high level of morbidity in the world [19,62,70]. As such, mRNA vaccines are a more secure alternative to traditional vaccine technology such as live or weak viruses and have made promising advances in the state of the art against viral infections. Examining the formulation properties of lipid nanoparticle–mRNA has enabled the rapid development and clinical use of these vaccines in the treatment of infectious diseases, cancer, and genetic disorders, as well as the protection and delivery of mRNA in vivo.

Nanotechnology and its dual delivery system can serve both as a carrier system and as an adjunct to RNA-based vaccination due to its similarity to microorganisms in structure and size. The nanosystem has the ability to increase the immune system response by simulating the natural infection process. In addition, it allows the encapsulation with safety stimuli to improve overall assist capacity. In the following sections, we examine some of the viral infectious diseases of the lungs and the effect of this system on them [26,27,44,60,63,71]. The evaluation of different clinical phases of lipid nanoparticle–mRNA vaccines for viral lung infections is described in Table 3.

### 7.1. Influenza

The first line of defense against influenza viruses is the innate immune system, which greatly increases damage in 650,000 people worldwide [72]. As the influenza virus mutates rapidly, causing antigenic drift, new strains are persistently emerging, with their characteristic drug resistance remaining a prevalent issue [6,14,26,72]. One of the key goals in this field is to create a broadly protective, universal influenza vaccine that can provide immunity against different strains of influenza and end seasonal vaccinations [71,73].

mRNA vaccines have a more effective treatment or prevention effect than conventional vaccines; through encoding the protected areas of the protein of the virus, it stimulates the production of specific antibodies [26]. One of the ways to control the influenza virus is to use nanoparticles, which, due to their aforementioned properties, have become excellent carriers of various antiviral agents [74]. The immunogenicity of mRNA-based vaccines assembled with lipid nanoparticles is much higher than other vaccines that are well tolerated in non-human mammals [75]. The intramuscular vaccination of non-human primates (NHPs) with LNP-formulated mRNAs encoding influenza antigens induced protective antibody titers that could remain stable for one year and become immunogenic [76]. Of course, mRNA vaccines can compete with inactivated virus-based vaccines, or even be important in terms of functional antibodies and T cell responses [75]. Compared to other vaccine platforms such as inactivated vaccines and recombinant proteins, GMP-grade lipid nanoparticle mRNA vaccines can be produced for specific antigens in a short time.

LNPs should be evaluated for safety and storage conditions prior to clinical use in vaccines. The mRNA-1440 and mRNA-1851 vaccines are examples of mRNA vaccines that are composed of lipid nanoparticles [60]. Vega et al. used a new platform that has high potential for mRNA vaccines called ASSET (Anchored Secondary scFv Enabling Targeting), which contains specific T cell monoclonal antibodies to target leukocytes to LNPs. Kranz et al. reported lipoplexes that target DCs after systemic delivery. The selective targeting of DC with mRNA vaccines is a critical finding for the stronger induction of immune responses [41]. Using a pure nucleoside-modified mRNA encapsulated in lipid nanoparticles (LNPs) that encode various viral surface antigens, Pradi et al. found that these vaccines were cells CD4 + Ts, which induce specific antigens and elicit strong plasma cell responses as well as neutralizing antibodies in mice and non-human mammals. They concluded that combining LNP technology with nucleoside modification was more effective [5,77].

In addition, the HA influenza virus from H10N8 and H7N9 elicits a strong protective immune response in mice and cynomolgus monkeys. These vaccines are also very effective in inducing the production of stable neutralizing antibodies, and the introduction of non-inflammatory modified nucleosides is essential for mRNA [5,77,78]. CureVac AG (Germany) has developed, for the first time, a sequence-engineering mRNA–lipid nanoparticle strategy to test the human flu seasonal virus RNA vaccine, enabling extraordinary levels of protein translation in vivo without the use of modified nucleoside [79]. Immunization with mRNA–lipid nanoparticles encoding HA from A/Hong Kong/4801/2014 (H3N2) also induces stronger immune responses of T and B cells than the MF59-licensed inactivated trivalent IAV vaccine [72]. In the case of influenza, the mRNA-1440 and mRNA-1851 vaccines, which include lipid nanoparticles and mRNA, have completed phase I clinical studies (NCT03076385 and NCT03314504). Hemagglutinin is an essential surface antigen of influenza viruses, which mRNA encodes from influenza viruses H10N8 and H7N9 (Figure 6) [60,64].

### 7.2. Coronavirus

Over the past two decades, there have been three significant occurrences of increased coronavirus infection—SARS-CoV, Middle East Respiratory Syndrome (MERS-CoV), and SARS-CoV-2 [80,81]. Although mRNA vaccines, with their rapid product development process, have played an important role in the development of vaccines, the SARS-CoV-2 pandemic is still ongoing, with efforts firmly focused on the generation and identification of novel therapeutic targets [20,42,82,83,84]. While the level of mRNA delivery is directly correlated to the level of antigen expression, repeat vaccination is required for current COVID-19 mRNA vaccines to achieve an adequate immune response [44,85]. With the onset of the pandemic, high-performance ionization lipids were reused for the COVID-19 mRNA vaccine formulation [42]. When the US-FDA authorized emergency use (EUA) for two vaccines, Moderna mRNA-1273 (Moderna) and BNT162b2 mRNA (Pfizer-BioNTech), it raised hopes of addressing the COVID-19 pandemic. For the first time in history, two mRNA-based vaccines have been developed using LNPs, lipid nanoparticles. Nanotechnology has contributed to the development of mRNA-based COVID-19 vaccines by Moderna and Pfizer/BioNTech. Currently, these two vaccines help diminish the COVID-19 pandemic and demonstrate the effective use of nanomedicine in combating health problems [21,42]. The success of these vaccines depends on the effective role of nanotechnology. In phase III clinical trials, it has been observed that the 95% success and efficacy of mRNA-based vaccines is due to the presence of their specific nanocarriers, LNPs [21,59,86,87].

One of the approaches that optimizes LNPs for COVID-19 vaccine delivery is the combination of DOPE and ionizing lipid–mRNA ratio, and LNP containing C12-200 is optimized for erythropoietin mRNA delivery and increases mRNA delivery efficiency [42]. BioNTech and Moderna mRNA vaccines use the LNP delivery system, which uses biodegradable ionizable lipids that form ester bonds in lipid tails (Figure 7). When the SM-102 lipid was used in the Moderna mRNA-1273 vaccine, it was found to perform better than Onpattro’s MC3 LNPs for intramuscular (IM) (Figure 8) mRNA delivery in rodents and non-human mammals [27]. COVID-19 mRNA vaccine injection is an intramuscular injection that elicits a transient thematic inflammatory response [44]. As an example, the mRNA-1273 (NCT04470427) and BNT162b2 (NCT04368728) vaccines both used ionizable lipid nanoparticles encoding SARS-CoV-2 full-length spike proteins (Figure 9) to deliver nucleoside-modified mRNAs [60,88]. In some human clinical studies, after the administration of SARS-CoV-2 mRNA-LNP vaccines, complications such as fever, pain, and swelling have been observed. However, due to the modified mRNA of these vaccines to reduce the identification and activation of innate immunity, these responses were due to the inflammatory part of the LNPs. Since the severe inflammatory responses of these mRNA-LNP vaccine platforms are related to the ionizable lipid component of LNPs, it is not yet clear whether these effects can be controlled by repeated injections that induce innate memory [86].

### 7.3. Respiratory Syncytial Virus (RSV)

Respiratory syncytial virus (RSV), with two subgroup, RSV A and RSV B, is the major cause of lower respiratory tract illness in infants of six months of age and less. Infants exposed to RSV in their respiratory system are at high risk of contracting the infection in childhood and adulthood, and current infections are at their highest-ever recorded rates. The significant side effects of RSV include respiratory distress, pneumonia, and bronchiolitis, which can be fatal. To date, there are no candidate vaccines approved for use in clinical applications [89,90]. RSV genome structure includes ten genes that express eleven proteins. RSV fusion F protein and glycoprotein G were discovered as the major antigens expressed, and the consequential target of neutralizing and protective antibodies (Figure 10). Recent studies describe a mechanism in which the humoral responses to natural RSV effects focus on targeting the prefusion complex [90].

An mRNA/LNP vaccine candidate for two RSV species (A and B) was developed that is shown to express prefusion. However, some obstacles still remain in identifying the optimal nanoparticle delivery system. There is currently an RSV vaccine candidate undergoing clinical evaluation [89]. The candidate vaccine is delivered by a custom protein nanomaterial used to produce a nanoparticle. This nanoparticle immunogenicity is ten-fold more powerful in neutralizing antibody responses than trimeric DS-Cav1 [89,91].

## 8. Using Particles as a Delivery System

### 8.1. LNPs as a Delivery System

Combining anti-fusion, anti-immune subversion, and anti-replication approaches by utilizing an NP system of phospholipid-coated chitosan NPs, will hopefully elicit a powerful immune purpose [92]. The mRNA-based vaccine that uses LNP for the delivery system improves the immune response to the vaccine, most likely through protecting the mRNA from enzymes and enabling efficient uptake and intracellular release of the mRNA in target cells [93].

### 8.2. VLPs as a Delivery System

Genetically engineered virus-like particles are an emerging tool within the field. VLPs are multiple duplicates of protein antigens in a particulate virus-like structure with a lack of viral genetic of the virus with limited pathogenicity and replication [94,95]. Eun-Ju Ko demonstrated a VLP-based vaccine that was developed and injected in a mouse model of RSV-F and RSV-G which had efficient immunogenicity without causing eosinophilia, and consequently significantly reduced viral replication within the lung [94].

## 9. Challenges

In general, although nanoparticle delivery systems have many advantages over other treatments, they still face several challenges, some of which may have toxicity or potential hazards to the system used. There are other challenges, such as targeted drug delivery and targeted drug release, to ensure the active components can be delivered to the right place in a limited time, and ensuring their biocompatibility [14,96]. Despite many rapid advances in vaccine technology, there are still challenges to mRNA vaccines [22]. One of the biggest challenges facing mRNA vaccines is the poor stability of mRNAs. Uncoated mRNA is rapidly degraded by extracellular RNases, and this mRNA can be immunogenic, so it alone cannot enter the cytosol for transcription. These two factors determine the need for intracellular mRNA delivery. So far, many methods for intracellular mRNA delivery have been studied, for example, the ex vivo loading of dendritic cells, physical delivery methods, cationic peptide protamine, and cationic LNPs delivery. Among these methods, the encapsulation and protection of mRNA in LNPs is a successful and attractive approach to this challenge [9,65]. The high molecular weight and negative charge of naked mRNA, its intrinsic instability, and its high susceptibility to ribonuclease degradation are obstacles to this pathway and make cell uptake difficult; for this reason, mRNA delivery systems have been developed which protect mRNA from degradation to provide better cellular uptake into target cells [63,97]. One approach to solving this problem is to create delivery systems based on cation carriers. These systems compress mRNA by encapsulating or complexing electrostatic interactions to produce nanoparticles capable of being adsorbed by the cell through endocytosis. It should be noted that if this interaction is very strong, mRNA is difficult to release in the cytosol to be translated, and the fit between the density and separation of mRNA from its carrier is an important and challenging issue. In the next step, these nanoparticles are released in the cytosol to prepare the mRNA for translation by separating it from its carrier [97,98,99]. Increased intracellular delivery resulting in endosomal escape is a successful delivery strategy in the recent endorsement of mRNA vaccines [99]. It should be noted that transfection efficiency improves with increasing endosomal escape. The crucial point for the development and design of next-generation nucleic acid vaccines is to consider various parameters during the vaccine manufacturing process, not only important for transfection efficiency but also for understanding the biological pathways in which transfected cells effectively enhance the protective immune response. In general, optimizing the safety profiles of NP formulations while maintaining their vaccine efficacy is also very significant [98]. The interesting point in recent reports is that LNP—mRNA is again encapsulated in extracellular vesicles (EV) and secreted from the receptor cells, which indicates that this exocytosis could be an alternative to producing nanocarriers (EV) for mRNA delivery [99,100]. Cholesterol and phospholipid components are less likely to develop inflammatory responses and innate immune recognition among LNP components because they are naturally present in mammalian cell membranes, but this is different for cationic/ionizable lipids because some of them cause inflammation by activating the TLR pathways and cytotoxicity. This is evidenced by the common side effects of inflammation, such as fever, swelling, and pain, in most people who have received SARS-CoV-2 mRNA-LNP-based vaccines [88]. Another challenge is the use of PEG polymers, which may produce anti-PEG antibodies and, in subsequent doses, create a phenomenon called accelerated blood clearance (ABC) [99]. Severe anaphylactic reactions have been reported following the administration of mRNA vaccines. On the other hand, PEGylated vaccines can cause anaphylactic reactions in people who already have high levels of anti-PEG antibodies, which is very dangerous. The US Centers for Disease Control and Prevention (CDC) emphasizes that people who are allergic to PEG (and polysorbate) should not receive the COVID-19 mRNA vaccine, and points out that it takes at least 15 min to access medical treatment for allergic reactions after all doses [99,101]. It should be noted that there is still little information on the structure and morphology of mRNA-formulated LNPs, the chemical stability of LNP components, and the colloidal stability of the mRNA-LNP system, as the effect of mRNA encapsulation in LNPs on the stability of mRNA vaccine storage is still unknown. As a result, more research is needed in this area [65].

## 10. Conclusions and Future Perspectives

Following the outbreak of COVID-19, the value of highly specific, quick-to-develop, and suitable for mass-scale manufacture vaccines quickly became apparent. With the rapid advances in vaccine technology over the last 3 years, the current state of the art has advanced significantly; however, much work yet remains to be conducted on both vaccine technology and vector/carrier technologies [102]. 

mRNA vaccines have now demonstrated significant potential as the go-to vaccine technology for viruses that can mutate to escape the immune system [59,67]. The use of LNPs was key to the development and optimization of these vaccines, which reduced mRNA degradation while targeting specific target cells for immunity [59]. The beneficial properties of nanoparticles such as biocompatibility, ability to encapsulate and modify the surface, targeted drug delivery, and the ability to deliver drugs slowly and stably make them a superior platform for the treatment and prevention of diseases compared to other conventional methods [14]. 

Much work is yet to be conducted on thoroughly addressing concerns around biocompatibility, toxicity, and other adverse side effects of the mRNA/nanoparticle approach, and new routes of administration, such as inhalation with nasal or lung targets, may be an opportunity to minimize dosages, whilst maximizing efficacy and raising the most appropriate immune response [87].

## Figures and Tables

**Figure 1 pharmaceutics-15-01127-f001:**
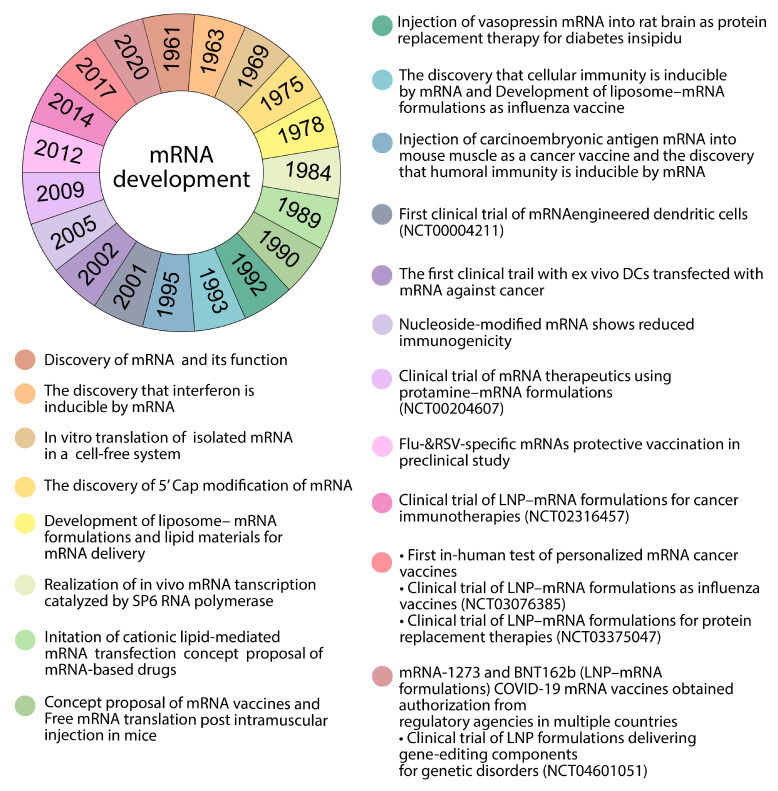
Timeline of selected significant events in mRNA development.

**Figure 2 pharmaceutics-15-01127-f002:**
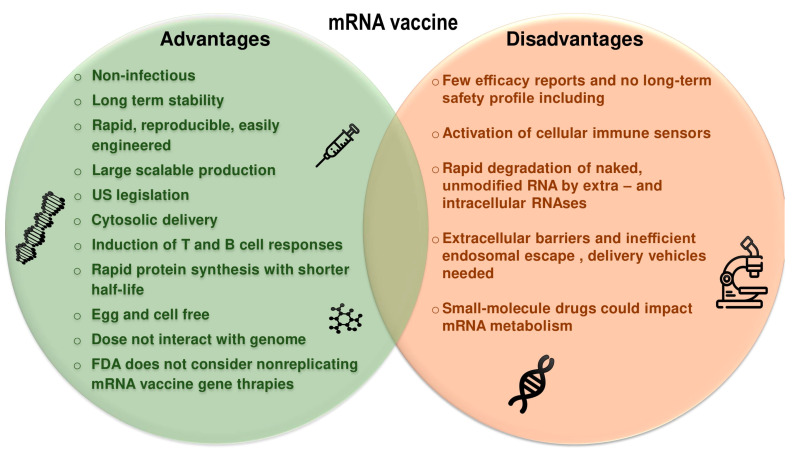
Comparison between the advantages and disadvantages of mRNA vaccines.

**Figure 3 pharmaceutics-15-01127-f003:**
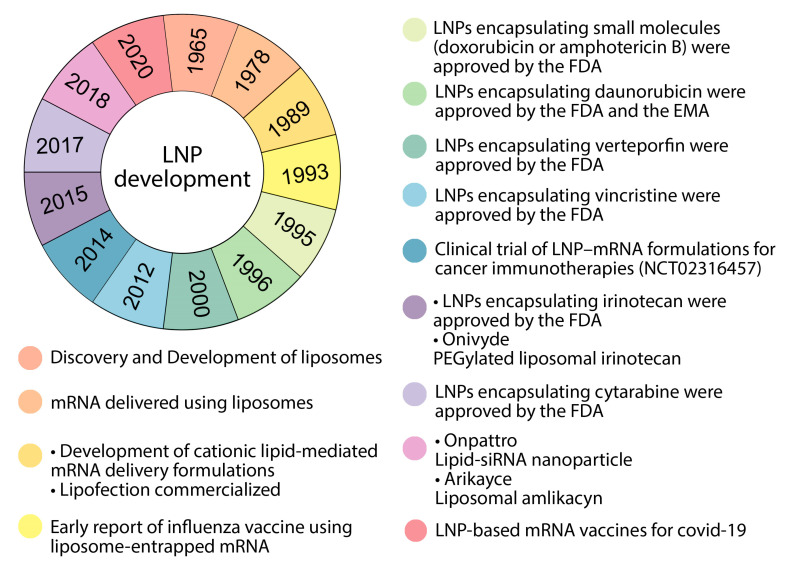
Timeline of some important events for LNP development. FDA: United States Food and Drug Administration, EMA: European Medicines Agency, LNP: lipid nanoparticle.

**Figure 4 pharmaceutics-15-01127-f004:**
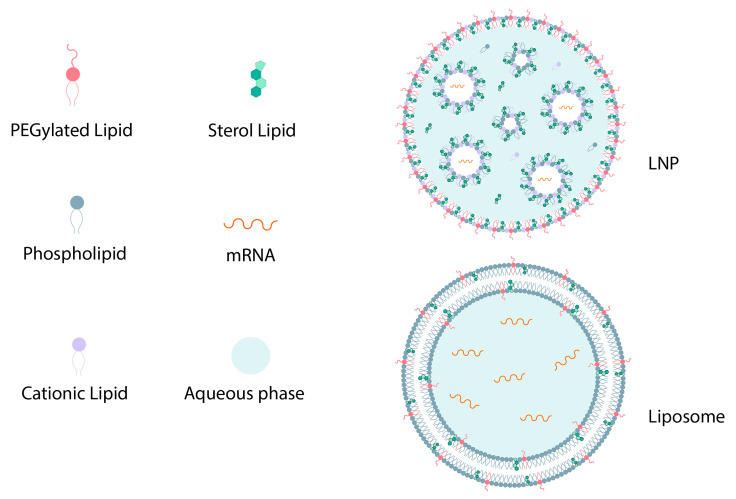
The general structure of an LNP and liposome.

**Figure 5 pharmaceutics-15-01127-f005:**
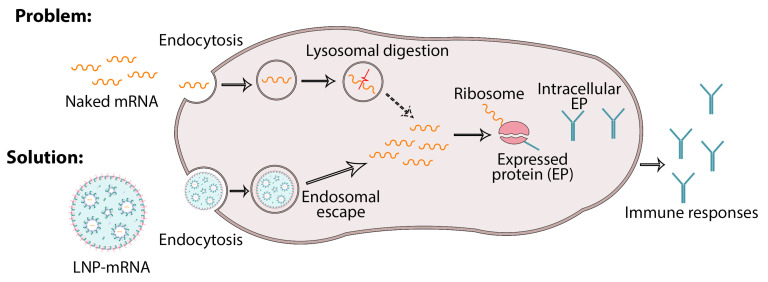
Adsorption of mRNA loaded in lipid nanoparticles into the host cells, which is a solution to prevent the degradation of naked mRNA in lysosomes. mRNA-loaded LNPs therefore undergo endosome escape and release mRNA for protein synthesis.

**Figure 6 pharmaceutics-15-01127-f006:**
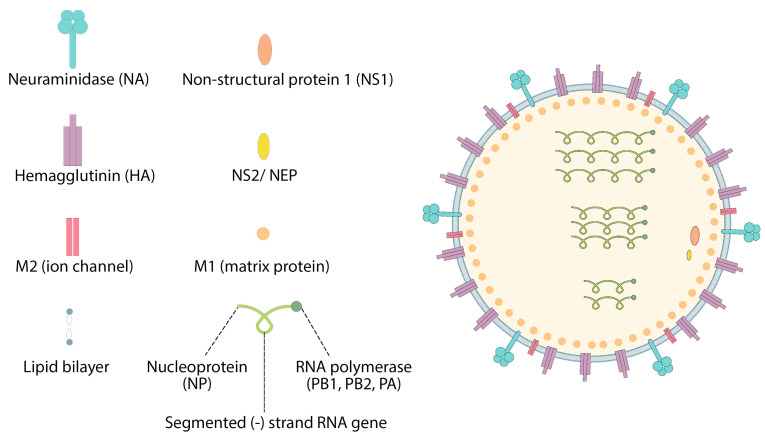
Structure of influenza virus.

**Figure 7 pharmaceutics-15-01127-f007:**
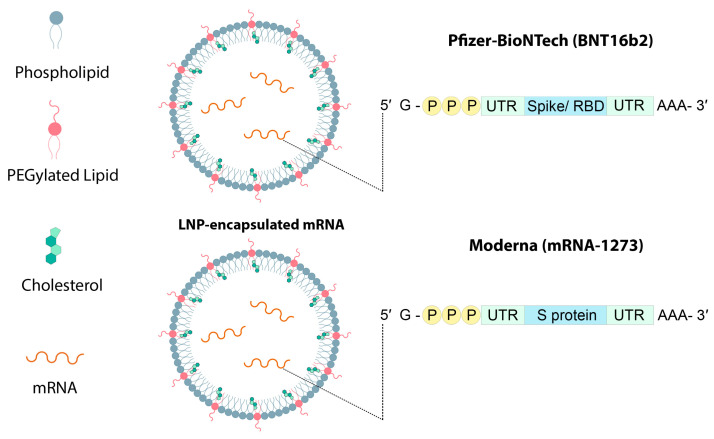
Structure of Pfizer-BioNTech (BNT162b2) and Moderna (mRNA-1273) vaccines.

**Figure 8 pharmaceutics-15-01127-f008:**
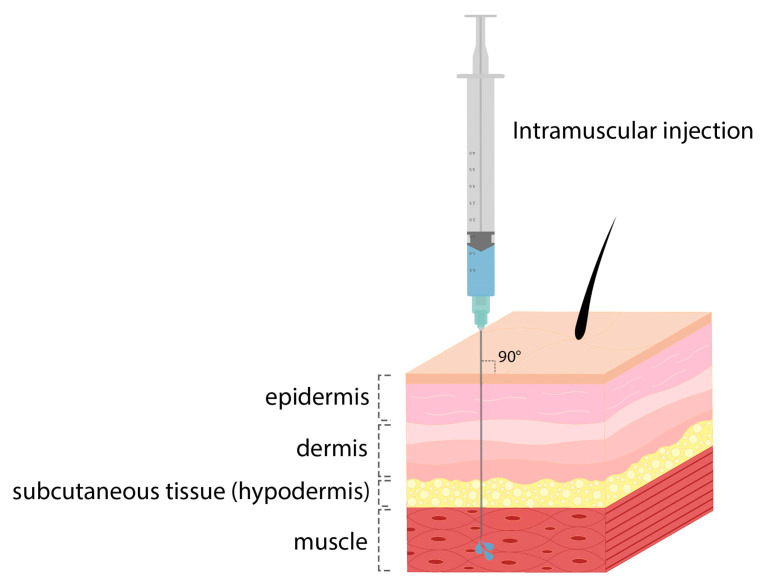
Intramuscular injection.

**Figure 9 pharmaceutics-15-01127-f009:**
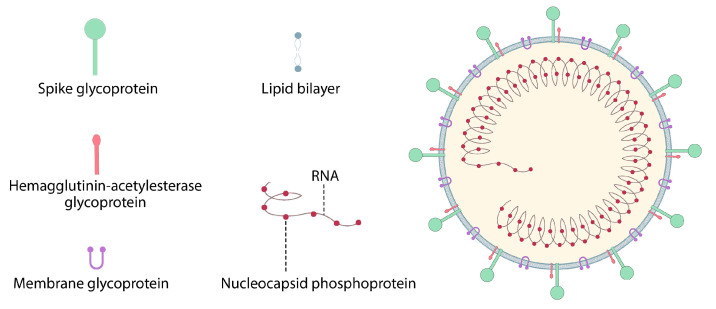
Structure of coronavirus.

**Figure 10 pharmaceutics-15-01127-f010:**
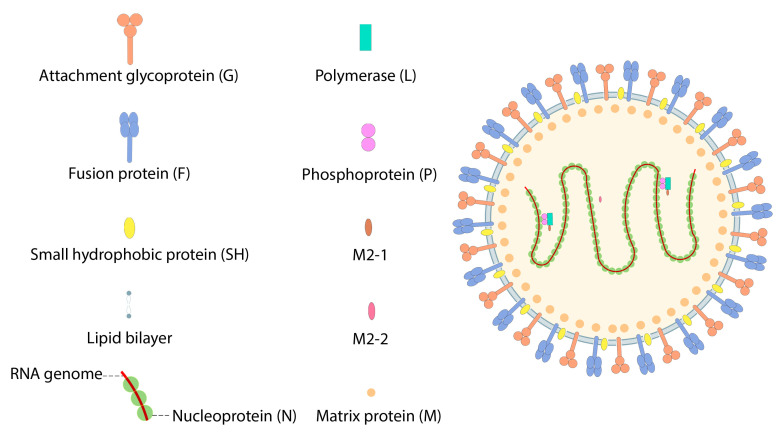
Structure of Respiratory Syncytial virus.

**Table 1 pharmaceutics-15-01127-t001:** The comparison between LNPs and liposomes.

Differences		Similarities
LNP	(1) Self-assembly (2) Phospholipid monolayer structure (3) The microfluidic preparation method	Shape, particle size distribution, positive charge, lipid composition
Liposome	(1) High energy dispersion (2) Phospholipid bilayer structure (3) Thin-film dispersion method	

**Table 2 pharmaceutics-15-01127-t002:** Recent lipid nanoparticle delivery systems for mRNA-based vaccines.

Indication	Nanodelivery System Compositions	Route of Administration	In Vivo Model
Influenza virus	Ionizable lipid, DSPC, cholesterol, PEG lipid	Intramuscular	Rodents and NHPs
Influenza virus	DOTAP, DOPE, (DSPE-Mpeg2000), Mannose	Intranasal	Mice
COVID-19	Ionizable lipid, DSPC, cholesterol, PEG lipid	Intramuscular	Mice and NHPs
Respiratory syncytial virus	Ionizable lipid, DSPC, cholesterol, PEG lipid	Intramuscular	Mice and cotton rats

NHPs; non-human primates.

**Table 3 pharmaceutics-15-01127-t003:** Clinical trials of lipid nanoparticle–mRNA vaccines for lung viral infectious diseases.

Name	Funding Source	Disease	Encoded Antigen	Administration Route	Stage	ClinicalTrials.gov Identifier
mRNA-1440 (nucleoside-modified)	Moderna Therapeutics	Influenza H10N8	Haemagglutinin	i.m.	Phase I	NCT03076385
mRNA-1851 (nucleoside-modified)	Moderna Therapeutics	Influenza H7N9	Haemagglutinin	i.m.	Phase I	NCT03345043
mRNA-1273 (perfusion-stabilized S protein mRNA vaccine)	Moderna Therapeutics/National Institute of Allergy and Infectious Diseases (NIAID)	COVID-19	Spike	i.m.	Phase III	NCT04470427
BNT162 (3 LNP-mRNA vaccines)	BioNTech/Pfizer	COVID-19	Spike	i.m.	Phase III	NCT04537949
BNT162b2	Pfizer	SARS-CoV-2	Spike	i.m.	Phase III (EUA and CMA)	NCT04368728
CVnCoV	CureVac AG	SARS-CoV-2	Spike	i.m.	Phase III	NCT04652102
ARCT-021	Arcturus Therapeutics	SARS-CoV-2	Spike	i.m.	Phase II	NCT04728347
mRNA-1345	Moderna	RSV	F glycoprotein	i.m.	Phase I	NCT04528719

EUA; emergency use authorization, CMA; conditional marketing authorization, i.m.; intramuscular, SARS-CoV-2; severe acute respiratory syndrome coronavirus 2, RSV; respiratory syncytial virus.

## Data Availability

Not applicable.

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
