# Peer review of "Lipid Nanoparticles as Promising Carriers for mRNA Vaccines for Viral Lung Infections"

_pharmaceutics, 2023, doi:10.3390/pharmaceutics15041127_

Round 1

Reviewer 1 Report

Due to the inherent characteristics of viral disease, which include complexity in the life cycle, different stages of proliferation in different chambers or subcellular organs, differences in replication dynamics, the possibility of latent infection and drug resistance, it has become necessary to develop new treatment modalities in the context of infection.

So why not synthesize encoded mRNA monoclonal antibodies? insert  lines for this technology. mRNA vaccines are not and will not solely be antigen – based.

it is also important to note that nanoparticles and nanotechnology can provide new solutions to some problems of traditional medicines and vaccines, such as insolubility in water, sensitivity, and absorption

How then will nanoparticles solve the issue of mRNA’s insolubility ? Is mRNA insoluble?

Vaccination and mechanism of mRNA

You mean mRNA vaccination because there are also other types?

How do traditional vaccines pose an infection risk to vaccinated persons, please elaborate.

transmit RNA ???

parts of the genome are copied 

Why parts of genome ?

Through modifying the mRNA sequence, vaccine production can be optimised.

Elaborate this statement

Although encoded antigens are different, most mRNA vaccine production and purification processes are quite similar – so there exist slight differences, elaborate.

 Comparison between the advantages and disadvantages?

Does mRNA production require bacterial cultures?

The main challenge for the development of mRNA in vaccines lies in the optimization of stability and delivery systems due to the instability and easy degradation of mRNA molecules

What is stability system?

this increases the thermal stability of the vaccine to allow storage and transport at room temperature – elaborate on this

LNP-based mRNA vaccines are effective and can be produced quickly

How quickly? Can they even be produced continuously?

most mRNA is destroyed you mean decomposed?

Through targeting these cell, this prolongs antigen production?

nanoparticle bodies?

against larger particles?

Nanoparticle drug delivery systems must have certain characteristics and principles?

on different drug poles?

Nanoplatforms that can be used to deliver mRNA in the body include polymer-based nanoparticles, polymer-lipid hybrid nanoparticles, polymer-based nanoparticles, polymer-lipid hybrid nanoparticles,

via Endosome

aqueous core (Kuntsche et al. , 2011),

Tables 1,2 unreadable

dual-delivery nano-delivery system?

Which transmits antigen against viral threats?

When the US-FDA authorized emergency use (EUA) for two vaccines, Moderna mRNA-1273 (Moderna) and BNT162b2 mRNA (Pfizer-BioNTech), it raised hopes of addressing the COVID-19 pandemic?

 Using some particles as a delivery system?

future administration of mRNA vaccines be replaced by intranasal

administration, which is a less invasive and easier route that will cause less discomfort

than intramuscular administration, or may even activate single-dose vaccines in the near

future will be available?

The letter sizes, fond, styles are incorrect, a lot of errors in expression and bibliographic terms are misused - epidermic approach on several chapters. Please go through the text from the beginning until the end. The conclusion part is very very weak, you must write it again.

Author Response

Thank you very much for having considered our manuscript " Nanodrug delivery systems; promising carriers for mRNA vaccines for viral lung infections" for publication in the Journal of Pharmaceutics. We appreciate to have received a positive evaluation, and we would like to express our appreciation to the Reviewers for the thoughtful comments and helpful suggestions. We have incorporated all of the suggestions made by the reviewers. Our detailed, point-by-point responses to reviewer comments are given in files, whereas the corresponding revisions are marked by using Track Changes in the manuscript file.

Reviewer 2 Report

Review to the manuscript pharmaceutics-2227387-peer-review-v1

This review summarized the nanodrug delivery systems, especially lipid nanoparticles (LNPs) , as promising carriers for mRNA vaccines for viral lung infections. Notably, mRNA vaccines are recognized as a new era in vaccination and play an important role in defeating COVID-19.

They also detailed the timeline of some important events for mRNA and LNPs development, respectively, and demonstrated the comparison between the advantages and disadvantages of mRNA vaccine, together expounded the components and delivery system of mRNA -LNP vaccines against viral lung infections.

Moreover, they provide a succinct overview of current challenges and potential future directions for nanodrug delivery systems with mRNA vaccines.

The manuscript put forward the important academic value and performs as accurate text expression, standard chart, good summary. Overall, this is a well-structured review, and suggested to be published directly after careful examination for possible minor errors.

Author Response

(The authors gave the same response as above.)

Reviewer 3 Report

The paper is interesting and focused on a very up-to date topic, giving a comprehensive overview of the state of the art in the field of nanodrug delivery stystems. In my opinion there are only few issues to be fixed before publication.

Figure 1 and 3: some points are expressed as bullet list and other no, please verify 

References in the main text: some references are not numbered but inserted in the form "author et al": citation style should be consistent thorughout the whole text. Please verify

Table 1: please verify headers alignment

Main text: there are some stylistic issues to be fix. Carefully proofread the text as in some sections there are sentences that are difficult to understand and other that seems to be incomplete.

Author Response

(The authors gave the same response as above.)
